# HYPERATTENTION: LONG-CONTEXT ATTENTION IN NEAR-LINEAR TIME

**Insu Han**
Yale University
insu.han@yale.edu

**Rajesh Jayaram**
Google Research
rkjayaram@google.com

**Amin Karbasi**
Yale University, Google Research
amin.karbasi@yale.edu

**Vahab Mirrokni**
Google Research
mirrokni@google.com

**David P. Woodruff**
CMU, Google Research
dwoodruf@cs.cmu.edu

**Amir Zandieh**
Independent Researcher
amir.zed512@gmail.com

## ABSTRACT

We present an approximate attention mechanism named "HyperAttention" to address the computational challenges posed by the growing complexity of long contexts used in Large Language Models (LLMs). Recent work suggests that in the worst-case scenario, quadratic time is necessary unless the entries of the attention matrix are bounded or the matrix has low stable rank. We introduce two parameters which measure: (1) the max column norm in the normalized attention matrix, and (2) the ratio of row norms in the unnormalized attention matrix after detecting and removing large entries. We use these fine-grained parameters to capture the hardness of the problem. Despite previous lower bounds, we are able to achieve a linear time sampling algorithm even when the matrix has unbounded entries or a large stable rank, provided the above parameters are small. HyperAttention features a modular design that easily accommodates integration of other fast low-level implementations, particularly FlashAttention. Empirically, employing Locality Sensitive Hashing (LSH) to identify large entries, HyperAttention outperforms existing methods, giving significant speed improvements compared to state-of-the-art solutions like FlashAttention. We validate the empirical performance of HyperAttention on a variety of different long-context length datasets. For example, HyperAttention makes the inference time of ChatGLM3 50% faster on 32k context length while perplexity increases from 5.6 to 6.3. On larger context length, e.g., 131k, with causal masking, HyperAttention offers 22-fold speedup on a single attention layer.

## 1 INTRODUCTION

Transformers (Vaswani et al., 2017) have been successfully applied to a wide variety of learning tasks in areas such as natural language processing (Devlin et al., 2018; Yang et al., 2019; Brown et al., 2020; Raffel et al., 2020), computer vision (Carion et al., 2020; Dosovitskiy et al., 2021), and time series forecasting (Zhou et al., 2021). Despite their success, these models face serious scalability limitations because naïve exact computation of their attention layers incurs quadratic (in the sequence length) runtime and memory complexities. This presents a fundamental challenge for scaling transformer models to longer context lengths.

Various approaches have been explored to tackle the quadratic-time attention layer, with one notable direction focusing on approximating intermediate matrices in attention layers. Methods for doing this include approximations by sparse matrices (Kitaev et al., 2020; Daras et al., 2020; Roy et al., 2021; Sun et al., 2021; Ding et al., 2023; Han et al., 2023), low-rank matrices (Choromanski et al., 2021; Katharopoulos et al., 2020; Kacham et al., 2023), or a combination of both (Chen et al., 2021b; Zaheer et al., 2020; Chen et al., 2021a; Dass et al., 2022). These methods aim to provide faster approximation to various components of attention, but none of them provide end-to-end approximations of the full dot-product attention. Moreover, none of these works support the use of causal masking, which is a crucial part of modern transformer architecture. In an ongoing work,

Kacham et al. (2023) bypass the hardness of softmax by switching to polynomial functions and applying fast polynomial sketching methods (Ahle et al., 2020; Woodruff & Zandieh, 2020; 2022). On the negative side, recent theoretical bounds suggest that entry-wise approximations to the attention matrix are impossible in sub-quadratic time in general (Alman & Song, 2023).

Nevertheless, recently, KDEFormer (Zandieh et al., 2023) has provided provable approximation in subquadratic time, under the assumption that the entries of the attention matrix are bounded. Theoretically, KDEFormer runs in roughly $\tilde{O}(n^{1.173})$ time; it employs *kernel density estimation* (KDE) to approximate column norms, allowing one to compute probabilities with which to sample columns of the attention matrix. However, the current algorithms for KDE are lacking practical efficiency (Charikar et al., 2020), and even in theory, there is a gap between the runtime of KDEFormer and the theoretically feasible $O(n)$ time algorithms. In (Alman & Song, 2023), the authors demonstrated that under the same assumption of bounded entries, a nearly linear time $O(n^{1+o(1)})$ algorithm is possible. However, their algorithm also involves using the polynomial method to approximate the softmax and is likely impractical (e.g., it was not empirically evaluated by the authors). In this work, we provide an algorithm which achieves the best of both worlds, being both a **(1)** practically efficient algorithm that **(2)** achieves the best possible near-linear time guarantee. Additionally, our approach supports casual masking, which was not possible via previous works.

## 1.1 PROBLEM STATEMENT

The *dot-product attention* (Vaswani et al., 2017) involves processing three input matrices: $\boldsymbol{Q}$ (queries), $\boldsymbol{K}$ (keys), $\boldsymbol{V}$ (values), all of size $n \times d$, where $n$ is the number of tokens in the input sequence and $d$ is the dimension of latent representations. This process outputs the following:

$$\mathbf{Att} = \boldsymbol{D}^{-1}\boldsymbol{A}\boldsymbol{V}$$

Here, matrix $\boldsymbol{A} := \exp\left(\boldsymbol{Q}\boldsymbol{K}^\top\right)$ is defined as the element-wise exponential of $\boldsymbol{Q}\boldsymbol{K}^\top$. Additionally, $\boldsymbol{D}$ is an $n \times n$ diagonal matrix derived from the sum of rows of $\boldsymbol{A}$, $\boldsymbol{D}_{i,i} = \|\boldsymbol{A}_{i,:}\|_1$ for $i \in [n]$. In this context, matrix $\boldsymbol{A}$ is referred to as the "attention matrix", and $\boldsymbol{D}^{-1}\boldsymbol{A}$ is called the "softmax matrix". It is important to note that calculating the attention matrix $\boldsymbol{A}$ directly requires $\Theta(n^2 d)$ operations, and storing it consumes $\Theta(n^2)$ memory. Consequently, a straightforward computation of $\mathbf{Att}$ demands a runtime of $\Omega(n^2 d)$ and $\Omega(n^2)$ memory.

Our objective is to efficiently approximate the output matrix $\mathbf{Att}$ while retaining its spectral properties. Our strategy involves designing an efficient estimator for the diagonal scaling matrix $\boldsymbol{D}$ in near-linear time. Additionally, we aim to quickly approximate the matrix product of the softmax matrix $\boldsymbol{D}^{-1}\boldsymbol{A}$ and value matrix $\boldsymbol{V}$ through subsampling. To be more specific, our objective is to find a sampling matrix $\boldsymbol{S} \in \mathbb{R}^{m \times n}$ with a limited number $m = n^{o(1)}$ of rows, along with a diagonal matrix $\widetilde{\boldsymbol{D}} \in \mathbb{R}^{n \times n}$, such that the following bound on the *operator norm* of the error is met:

$$\left\|\mathbf{Att} - \widetilde{\boldsymbol{D}}^{-1}\boldsymbol{A}\boldsymbol{S}^\top \cdot \boldsymbol{S}\boldsymbol{V}\right\|_{\mathrm{op}} \leq \varepsilon \cdot \left\|\boldsymbol{D}^{-1}\boldsymbol{A}\right\|_{\mathrm{op}} \|\boldsymbol{V}\|_{\mathrm{op}}. \tag{1}$$

## 1.2 OUR CONTRIBUTIONS

We show that efficiently solving the matrix multiplication component of the attention approximation problem in Eq. (1) can be achieved by defining the sampling matrix $\boldsymbol{S}$ based on the row norms of $\boldsymbol{V}$. The more challenging aspect lies in obtaining a reliable spectral approximation for the diagonal matrix $\boldsymbol{D}$. In a recent result, Zandieh et al. (2023) effectively leverages fast KDE solvers to attain a high-quality approximation of $\boldsymbol{D}$. However, we streamline the KDEformer procedure and demonstrate that uniform sampling is sufficient to achieve the desired spectral guarantee, eliminating the need for importance sampling based on kernel densities. This significant simplification allows us to develop a practical *and* provably linear time algorithm.

In contrast to prior work (Alman & Song, 2023; Zandieh et al., 2023), our approach does not necessitate bounded entries or bounded stable rank. Furthermore, the fine-grained parameters we introduce to analyze the time complexity may remain small even when the entries in the attention matrix or the stable rank are large.

Our work is inspired by the hard instance of Alman & Song (2023) for showing quadratic time lower bounds. Such instances have one randomly placed large entry in each row of the attention matrix. Our algorithm has an initial phase where we find large entries of the attention matrix in a black box manner, such as by using Locality Sensitive Hashing (Kitaev et al., 2020), or a possibly learned CountSketch applied to the attention matrix (Charikar et al., 2002; Li et al., 2023a), or just a known heavy entry pattern (Chen et al., 2021a). We assume these procedures are fast, and that after removing the heavy entries, two parameters in the resulting attention matrix are small: (1) the max column $\ell_2$-norm, and (2) the ratio of row norms in the un-normalized attention matrix.

Prior work of Zandieh et al. (2023) used KDE to identify columns in the attention matrix with large norm and to perform approximate matrix product with the value matrix by sampling such columns. As mentioned, finding such columns requires at least $O(n^{1.173})$ time. Instead, we observe that by doing a one-sided sampling from the squared row norms of $V$, we can avoid the use of KDEs and achieve the same spectral norm guarantee in terms of the stable rank. Although our algorithm is simple and just samples by the row norms of the value matrix (or even samples uniformly in practice), the main technical challenge is that we do not know the row norms of the attention matrix needed in order to normalize it and produce a proper factorization of it. This is reminiscent of the quadratic time hard instance of (Alman & Song, 2023) where we may not be able to find a heavy entry in a row easily, and thus cannot normalize by its norm in the attention matrix. Our parameters (1) and (2) above allow us to argue that the heavy entries, if they exist, are not distributed in the worst possible way.

Empirically, HyperAttention demonstrates significant speed improvements, achieving over a $50\times$ acceleration in forward and backward propagation for sequence lengths of $n = 131\text{k}$. When dealing with causal masking, the method still delivers a substantial $5\times$ speedup. Moreover, when our approach is applied to pretrained LLMs, e.g., `chatglm3-6b-32k` (Du et al., 2022) and evaluated on long-context benchmark datasets, so-called LongBench (Bai et al., 2023), it maintains performance levels that closely match those of the original models, even without the need for fine-tuning. Furthermore, we investigate task-specific evaluations and discover summarization and code completion tasks are more robust to approximate attention layers than question answerings.

## 2 PRELIMINARIES

We make use of the *Hamming sorted LSH*, a variant of angular locality-sensitive hashing introduced in the work by Zandieh et al. (2023). In this variant, the hash buckets are arranged in order of their Hamming distances. This LSH variant is particularly well-suited for designing GPU-friendly algorithms aimed at identifying dominant entries within the attention matrix $A$. In the context of Hamming sorted LSH, if we let $\mathcal{H} : \mathbb{R}^d \to [B]$ be a hash function with $B$ buckets drawn from an LSH family, then the collision probability $\Pr_{\mathcal{H}}[\mathcal{H}(q) = \mathcal{H}(k)]$ is "roughly" proportional to $\langle q, k \rangle$. A very useful property of this LSH variant is that its buckets are ordered in such a way that geometrically adjacent buckets have consecutive buckets. We provide the following definition.

**Definition 1** (Hamming sorted LSH, Definition 7.3 of (Zandieh et al., 2023)). *For positive integer $r$, there exists an LSH function $\mathcal{H} : \mathbb{R}^d \to [2^r]$, such that for any $x, y \in \mathbb{R}^d$ its collision probability is $\Pr[\mathcal{H}(x) = \mathcal{H}(y)] = \left(1 - \frac{\theta(x,y)}{\pi}\right)^r$ where $\theta(x,y) := \cos^{-1}\left(\frac{x^\top y}{\|x\|\|y\|}\right)$. Furthermore, this LSH function hashes similar points to adjacent buckets. Specifically, the probability that two points end up in adjacent buckets is given by $\Pr\left[\mathcal{H}(x) = \mathcal{H}(y) \pm 1 \pmod{2^r}\right] = \frac{2\theta(x,y)}{\pi} \cdot \left(1 - \frac{\theta(x,y)}{\pi}\right)^{r-1}$.*

Using this LSH function, as demonstrated by Zandieh et al. (2023), we can sort keys and queries within an attention layer in such a way that large entries get shifted towards the diagonal of the attention matrix. Subsequently, these significant entries in the attention matrix can be captured by computing equal-sized blocks along the diagonal. This approach aligns with the block-memory access patterns of modern hardware and can be efficiently parallelized through batching across blocks.

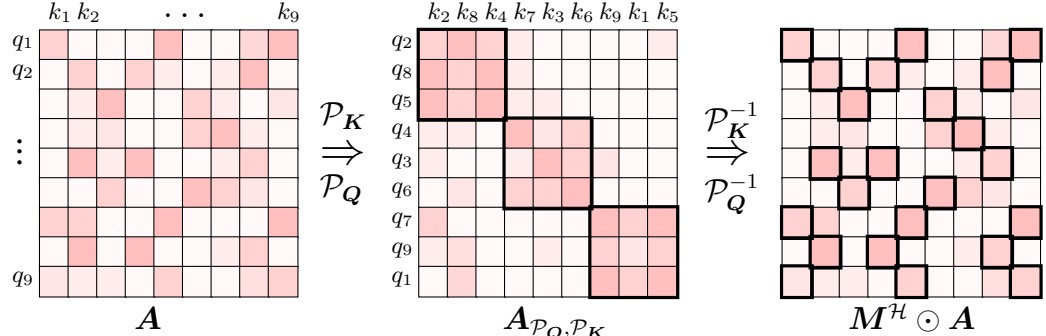

Figure 1: How sortLSH finds large entries of $\boldsymbol{A}$: (Left) Keys and queries undergo hashing using the Hamming ordered LSH $\mathcal{H}(\cdot)$. (Middle) Keys and queries are rearranged based on their hash buckets. Attention matrix after applying these row and column permutations is denoted as $\boldsymbol{A}_{\mathcal{P}_{\boldsymbol{Q}}, \mathcal{P}_{\boldsymbol{K}}}$. Large entries of $\boldsymbol{A}_{\mathcal{P}_{\boldsymbol{Q}}, \mathcal{P}_{\boldsymbol{K}}}$ are concentrated around the diagonal blocks. (Right) rows and columns permutations are reversed on the attention matrix and $\boldsymbol{M}^{\mathcal{H}} \odot \boldsymbol{A}$ is highlighted.

---

**Algorithm 1:** sortLSH for locating large entries of $\boldsymbol{A}$

---

1: **input**: matrices $\boldsymbol{Q}, \boldsymbol{K} \in \mathbb{R}^{n \times d}$, and block size $b$
2: Let $\mathcal{H}(\cdot)$ be a Hamming sorted LSH as per Definition 1 and hash rows of $\boldsymbol{Q}, \boldsymbol{K}$
3: Let $\mathcal{P}_{\boldsymbol{K}}, \mathcal{P}_{\boldsymbol{Q}} \in \mathrm{Sym}(n)$ be permutations satisfying $\mathcal{P}_{\boldsymbol{K}}(i) < \mathcal{P}_{\boldsymbol{K}}(j)$ if $\mathcal{H}(\boldsymbol{K}_{i,:}) \leq \mathcal{H}(\boldsymbol{K}_{j,:})$ and $\mathcal{P}_{\boldsymbol{Q}}(i) < \mathcal{P}_{\boldsymbol{Q}}(j)$ if $\mathcal{H}(\boldsymbol{Q}_{i,:}) \leq \mathcal{H}(\boldsymbol{Q}_{j,:})$
4: **return** Mask matrix $\boldsymbol{M}^{\mathcal{H}} \in \{0,1\}^{n \times n}$ defined as $\boldsymbol{M}_{i,j}^{\mathcal{H}} = \mathbf{1}_{\{\lfloor \mathcal{P}_{\boldsymbol{Q}}(i)/b \rfloor = \lfloor \mathcal{P}_{\boldsymbol{K}}(j)/b \rfloor\}}$

---

## 3 ALGORITHM

To obtain a spectral guarantee when approximating $\mathbf{Att}$, our initial step involves producing a $1 \pm \varepsilon$ approximation of the diagonal entries in the matrix $\boldsymbol{D}$. Subsequently, we approximate the matrix product between $\boldsymbol{D}^{-1}\boldsymbol{A}$ and $\boldsymbol{V}$ via sampling according to the squared row $\ell_2$-norms of $\boldsymbol{V}$.

**Estimating $\boldsymbol{D}$.** Our procedure for approximating $\boldsymbol{D}$ consists of two steps. Initially, we identify the dominant entries within the attention matrix using an algorithm rooted in the Hamming sorted LSH, as defined in Definition 1. The second step revolves around randomly selecting a small subset of keys $\boldsymbol{K}$. We will demonstrate that under certain mild assumptions about matrices $\boldsymbol{A}$ and $\boldsymbol{D}$, this simple approach allows us to establish spectral bounds on the estimated matrix. Our aim is to find a sufficiently precise approximate matrix $\widetilde{\boldsymbol{D}}$ that satisfies:

$$\left\| \left( \widetilde{\boldsymbol{D}}^{-1} - \boldsymbol{D}^{-1} \right) \boldsymbol{A} \right\|_{\mathrm{op}} \leq \frac{\varepsilon}{2} \left\| \boldsymbol{D}^{-1} \boldsymbol{A} \right\|_{\mathrm{op}} \tag{2}$$

Our assumption is that the column norms of the softmax matrix exhibit a relatively uniform distribution. To be more precise, we assume that for any $i \in [n]$ there exists some $\alpha = n^{o(1)}$ such that $\left\| \boldsymbol{D}^{-1}\boldsymbol{A} \cdot e^{(i)} \right\|_2^2 \leq \frac{\alpha}{n}$. It's worth noting that our assumption is more general in comparison to the bounded input entries assumption made in (Alman & Song, 2023). In fact, if their assumption holds, it implies that $\left\| \boldsymbol{D}^{-1}\boldsymbol{A} \cdot e^{(i)} \right\|_2^2 \leq \frac{n^{o(1)}}{n}$ for all $i \in [n]$. In Section 4.3, we empirically compute $\alpha$ to be the maximum of the squared $\ell_2$-norms of the columns in $\boldsymbol{D}^{-1}\boldsymbol{A}$ and verify that it is indeed sublinear in $n$.

The first step of our empirical algorithm involves identifying large entries of the attention matrix $\boldsymbol{A}$ through hashing keys and queries into uniformly-sized buckets using the Hamming sorted LSH, which we refer to as *sortLSH*. This process is detailed in Algorithm 1 and is visually illustrated in Fig. 1. Note that we also mention other was of identifying large patterns, such as checking for a known heavy hitter pattern, or using CountSketch which we describe more below.

Algorithm 1 returns a sparse mask designed to isolate the dominant entries of the attention matrix. Given this mask, we compute an approximation of the matrix $\mathbf{D}$ in Algorithm 2 that satisfies the spectral guarantee in Eq. (2). This algorithm accomplishes this by combining the attention values corresponding to the mask with a randomly chosen subset of columns from the attention matrix.

---

**Algorithm 2:** ApproxD for estimating diagonal matrix $\boldsymbol{D}$

1: **input**: matrices $\boldsymbol{Q}, \boldsymbol{K} \in \mathbb{R}^{n \times d}$, large entries mask $\boldsymbol{M}^{\mathcal{H}} \in \{0, 1\}^{n \times n}$, parameters $\kappa > 0$, $\varepsilon > \frac{1}{\kappa^4}$, $\alpha > \varepsilon^2 \kappa$, and integer $m$
2: Randomly choose a subset $\mathbb{T} \subseteq [n]$ with cardinality $|\mathbb{T}| = m$
3: $\tau \leftarrow \max_{j \in \mathbb{T}} \left\langle 1 - \boldsymbol{M}_{j,:}^{\mathcal{H}}, \exp(\boldsymbol{K}\boldsymbol{Q}_{j,:}^{\top}) \right\rangle$     {estimate of maximum un-masked row sum}
4: Generate $m$ i.i.d. sample $\ell_1, \ell_2, \ldots \ell_m \sim \mathrm{Unif}([n])$
5: **for** $i \in [n]$ **do**
6:     $C_i \leftarrow \Theta\left(\frac{\varepsilon^2 m}{n \log n} \cdot \left(\langle \boldsymbol{M}_{i,:}^{\mathcal{H}}, \exp(\boldsymbol{K}\boldsymbol{Q}_{i,:}^{\top})\rangle + \tau/\kappa\right)\right)$     {(capped) row-sum of masked entries}
    {$d_i$ below is the (capped) row-sum estimator for the un-masked entries}
7:     $d_i \leftarrow \frac{n}{m} \sum_{j \in [m]} (1 - \boldsymbol{M}_{i,\ell_j}^{\mathcal{H}}) \cdot \min\left(\exp\left(\langle \boldsymbol{Q}_{i,:}, \boldsymbol{K}_{\ell_j,:}\rangle\right), C_i\right)$
8:     $\tilde{d}_i \leftarrow \langle \boldsymbol{M}_{i,:}^{\mathcal{H}}, \exp(\boldsymbol{K}\boldsymbol{Q}_{i,:}^{\top})\rangle + \max(d_i, \tau/\kappa)$     {full row-sum estimate}
9: **return** diagonal matrix $\widetilde{\boldsymbol{D}} = \mathrm{diag}(\{\tilde{d}_i\}_{i=1}^n)$

---

**Algorithm 3:** HyperAttention: attention mechanism in near-linear time

1: **input**: matrices $\boldsymbol{Q}, \boldsymbol{K}, \boldsymbol{V} \in \mathbb{R}^{n \times d}$, mask matrix $\boldsymbol{M}^{\mathcal{H}} \in \{0, 1\}^{n \times n}$, and parameter $\varepsilon > \frac{1}{n^{o(1)}}$
2: Run Algorithm 2 and let $\widetilde{\boldsymbol{D}} \leftarrow \textsc{ApproxD}\left(\boldsymbol{Q}, \boldsymbol{K}, \boldsymbol{M}^{\mathcal{H}}, n^{o(1)}, \varepsilon, n^{o(1)}, d \cdot n^{o(1)}\right)$
3: Let $\boldsymbol{S} \in \mathbb{R}^{m \times n}$ be an i.i.d. sampling matrix based on squared row norms of $\boldsymbol{V}$ as in Lemma 2
4: **return** $\widetilde{\boldsymbol{D}}$ and $\boldsymbol{S}$

---

The assumptions of Lemma 1 are used to ensure that the variance of the estimator is small, and the same complexity of the algorithm increases as a function of the parameters $\alpha, \kappa$. We remark that our algorithm is versatile and can function effectively with a predefined mask that specifies the positions of dominant entries within the attention matrix, mirroring the approach taken in (Chen et al., 2021a). The main guarantee provided by this algorithm is given in Lemma 1.

**Lemma 1** (Approximating $\boldsymbol{D}$). *For any $\boldsymbol{Q}, \boldsymbol{K} \in \mathbb{R}^{n \times d}$, let $\boldsymbol{A} = \exp(\boldsymbol{Q}\boldsymbol{K}^{\top})$. Also let $\boldsymbol{D} \in \mathbb{R}^{n \times n}$ be the diagonal matrix with $\boldsymbol{D}_{i,i} = \|\boldsymbol{A}_{i,:}\|_1$. Additionally, suppose that $\alpha = n \cdot \max_{i \in [n]} \left\|\boldsymbol{D}^{-1}\boldsymbol{A} \cdot e^{(i)}\right\|_2^2$. For any mask matrix $\boldsymbol{M}^{\mathcal{H}} \in \{0, 1\}^{n \times n}$ let us define the condition number $\kappa := \frac{\max_{i \in [n]} \langle 1 - \boldsymbol{M}_{i,:}^{\mathcal{H}}, \boldsymbol{A}_{i,:}\rangle}{\min_{j \in [n]} \langle 1 - \boldsymbol{M}_{j,:}^{\mathcal{H}}, \boldsymbol{A}_{j,:}\rangle}$. If $m = \Omega\left(\frac{\kappa^7 \cdot \alpha^2}{\varepsilon^6} \log n\right)$, the output $\widetilde{\boldsymbol{D}}$ of Algorithm 2 satisfies Eq. (2) with probability at least $1 - \frac{1}{\mathrm{poly}(n)}$.*

**Approximating the product of softmax matrix $\boldsymbol{D}^{-1}\boldsymbol{A}$ and values matrix $\boldsymbol{V}$.** Given a $\widetilde{\boldsymbol{D}}$ that meets the spectral approximation conditions as in Eq. (2), we can achieve the spectral constraint in Eq. (1), by finding a sampling matrix that satisfies the following condition,

$$\left\|\widetilde{\boldsymbol{D}}^{-1}\boldsymbol{A}\boldsymbol{S}^{\top} \cdot \boldsymbol{S}\boldsymbol{V} - \widetilde{\boldsymbol{D}}^{-1}\boldsymbol{A}\boldsymbol{V}\right\|_{\mathrm{op}} \leq \frac{\varepsilon}{2} \cdot \left\|\boldsymbol{D}^{-1}\boldsymbol{A}\right\|_{\mathrm{op}} \|\boldsymbol{V}\|_{\mathrm{op}} \tag{3}$$

We can efficiently find a sampling matrix $\boldsymbol{S} \in \mathbb{R}^{m \times n}$ with a small number $m$ of rows that satisfies Eq. (3) by leveraging well-established techniques in *Approximate Matrix Multiplication* (AMM).

**Lemma 2.** *For any matrices $\widetilde{\boldsymbol{D}}, \boldsymbol{A} \in \mathbb{R}^{n \times n}, \boldsymbol{V} \in \mathbb{R}^{n \times d}$ consider a sampling matrix $\boldsymbol{S} \in \mathbb{R}^{m \times n}$ constructed as follows: first generate $m$ i.i.d. samples $\ell_1, \ldots \ell_m \in [n]$ according to squared row norms of matrix $\boldsymbol{V}$, i.e., $\frac{\|\boldsymbol{V}_{i,:}\|_2^2}{\|\boldsymbol{V}\|_F^2}$, then let the $r^{th}$ row of $\boldsymbol{S}$ be $\frac{\|\boldsymbol{V}\|_F}{\sqrt{m} \cdot \|\boldsymbol{V}_{\ell_r,:}\|_2} \cdot e^{(\ell_r)}$. If $m = \Omega\left(\varepsilon^{-2}d \cdot \mathrm{srank}(\widetilde{\boldsymbol{D}}^{-1}\boldsymbol{A})\right)$ for some $\varepsilon > 0$, the following holds with probability at least 0.99:*

$$\left\|\widetilde{\boldsymbol{D}}^{-1}\boldsymbol{A}\boldsymbol{S}^{\top} \cdot \boldsymbol{S}\boldsymbol{V} - \widetilde{\boldsymbol{D}}^{-1}\boldsymbol{A}\boldsymbol{V}\right\|_{\mathrm{op}} \leq \varepsilon \cdot \left\|\widetilde{\boldsymbol{D}}^{-1}\boldsymbol{A}\right\|_{\mathrm{op}} \|\boldsymbol{V}\|_{\mathrm{op}}.$$

The above result is standard and for proof refer to (Drineas & Kannan, 2001).

**Main Theorem.** Now, we can integrate the subroutines for approximating the diagonal $\widetilde{\boldsymbol{D}}$ and approximating the matrix product between $\widetilde{\boldsymbol{D}}^{-1}\boldsymbol{A}$ and values matrix $\boldsymbol{V}$. With this, we introduce the

*HyperAttention*, an efficient algorithm that can approximate the attention mechanism with spectral guarantees as per Eq. (1) in near-linear time. Our Algorithm 3 takes as input a mask $M^{\mathcal{H}}$ that defines the positions of dominant entries within the attention matrix. This mask can be generated using the sortLSH algorithm (Algorithm 1), or it can be a predefined mask similar to the approach taken in (Chen et al., 2021a). The large entries mask $M^{\mathcal{H}}$ is assumed to be sparse by design and its number of nonzero entries is bounded $\text{nnz}(M^{\mathcal{H}}) = n^{1+o(1)}$. We now introduce our main theorem which will be proved in Appendix A.

**Theorem 1** (HyperAttention guarantee). *For any matrices $Q, K, V \in \mathbb{R}^{n \times d}$, any mask matrix $M^{\mathcal{H}} \in \{0, 1\}^{n \times n}$, and parameter $\varepsilon > \frac{1}{n^{o(1)}}$, let $A = \exp(QK^\top)$ and let $D \in \mathbb{R}^{n \times n}$ be the diagonal matrix with $D_{i,i} = \|A_{i,:}\|_1$. If $\max_{i \in [n]} \left\| D^{-1} A \cdot e^{(i)} \right\|_2^2 \leq \frac{n^{o(1)}}{n}$ and $\frac{\max_{i \in [n]} \langle 1 - M^{\mathcal{H}}_{i,:}, A_{i,:} \rangle}{\min_{j \in [n]} \langle 1 - M^{\mathcal{H}}_{j,:}, A_{j,:} \rangle} \leq n^{o(1)}$ then with probability at least $0.98$ the outputs $S, \widetilde{D}$ of Algorithm 3 satisfy the spectral condition as in Eq. (1). Moreover, this algorithm's runtime is $O(d \cdot n^{1+o(1)} + d \cdot \text{nnz}(M^{\mathcal{H}}))$.*

Note that even if $M^{\mathcal{H}}$ is not given to us, but $M^{\mathcal{H}}$ can be found in $d \cdot n^{1+o(1)}$ time, the theorem holds. We also give examples when this is possible by using Hamming sorted LSH, which our experiments are based on, or using the ExpanderSketch of (Larsen et al., 2016) which is based on CountSketch (Charikar et al., 2002) but also gives a fast recovery time. In the supplementary we show:

**Corollary 1** (HyperAttention with sortLSH). *Suppose all preconditions of Theorem 1 hold. Further, suppose the mask matrix $M^{\mathcal{H}} \in \{0, 1\}^{n \times n}$ is generated as in Algorithm 1 with block size $b = n^{o(1)}$ and $r = \log_2 n$ in Definition 1. We further assume there are at most $n^{1+o(1)}$ pairs $(i, j)$ with $\theta(Q_{i,*}, K_{j,*}) \leq \frac{\pi}{2}(1 - o(1))$, where $\theta$ is as in Definition 1. Then with probability $1 - 1/n^{o(1)}$, the $M^{\mathcal{H}}$ we find in Algorithm 1 has at most $n^{1+o(1)}$ non-zero entries and with probability at least .98, the outputs $S, \widetilde{D}$ of Algorithm 3 satisfy Eq. (1) and the overall runtime is $O(d \cdot n^{1+o(1)})$.*

We note the assumption on the angles of the rows of $Q$ and $K$ in Corollary 1 is satisfied if most rows are drawn uniformly at random from a $d$-dimensional sphere, since in this case they will be nearly orthogonal, i.e., have angle at most $\frac{\pi}{2}(1 - o(1))$ with high probability. However, the corollary also allows $n^{1+o(1)}$ pairs of rows to have arbitrary angle, which may be more realistic.

**Corollary 2** (HyperAttention with ExpanderSketch). *Suppose all preconditions of Theorem 1 hold. Further, suppose the mask matrix $M^{\mathcal{H}} \in \{0, 1\}^{n \times n}$ is defined such that there is a threshold $\tau = n^{o(1)}$ such that $M^{\mathcal{H}}_{i,j} = 1$ if and only if $(QK^\top)^2_{i,j} \geq \frac{\|QK^\top e_j\|_2^2}{\tau}$. Then we can find $M^{\mathcal{H}}$ exactly with probability $1 - O(1/n^2)$, and with probability at least .98, the outputs $S, \widetilde{D}$ of Algorithm 3 satisfy Eq. (1). The runtime is $O(d \cdot n^{1+o(1)})$.*

The key idea behind the proof of Corollary 2 is to first sketch $Q$ by an ExpanderSketch $T$, which is efficient since $T$ has a small number of rows. Then compute $(T \cdot Q) \cdot K^\top$ which is again efficient since $(T \cdot Q)$ has a small number of rows. Thus, we never form the matrix $Q \cdot K^\top$.

## 3.1 CAUSAL MASKING

Language models commonly employ causal masking. The causal mask is a lower triangular binary square matrix denoted as $M^{\mathcal{C}}$ where $M^{\mathcal{C}}_{i,j} = \mathbf{1}_{\{i \geq j\}}$. The causal attention mechanism is defined as:

$$\text{Att}_{\mathcal{C}} = D_{\mathcal{C}}^{-1}(M^{\mathcal{C}} \odot A)V,$$

where $A := \exp\left(QK^\top\right)$ is defined as before and $D_{\mathcal{C}}$ is an $n \times n$ diagonal matrix derived from the sum of rows of the masked attention $M^{\mathcal{C}} \odot A$, specifically $[D_{\mathcal{C}}]_{i,i} = \langle M^{\mathcal{C}}_{i,:}, A_{i,:} \rangle$ for $i \in [n]$. To approximate causal attention with a spectral guarantee, we require two components. First, we need a spectral approximation for the diagonal matrix $D_{\mathcal{C}}$. Second, we need to approximate the matrix product between $D_{\mathcal{C}}^{-1}(M^{\mathcal{C}} \odot A)$ and $V$, which can be achieved using the same sampling technique as described in Algorithm 3 and Lemma 2. The first component is more intricate, and we employ a recursive method to address it. So we focus on how to efficiently approximate the diagonal $D_{\mathcal{C}}$.

Our approach is based on a key observation, as depicted in Fig. 2. The masked attention $M^{\mathcal{C}} \odot A$ can be decomposed into three non-zero matrices, each of which has half the size of the original attention matrix. The block $A_{21}$, located entirely below the diagonal is unmasked attention. Consequently,

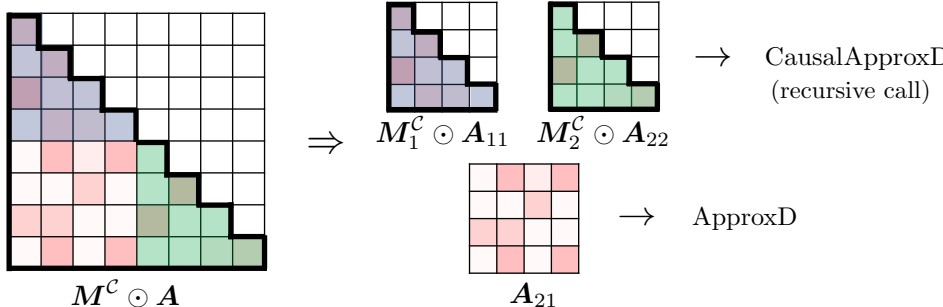

Figure 2: Causal attention matrix can be divided into three equal-sized non-zero sections: $M_1^{\mathcal{C}} \odot A_{11}$ and $M_2^{\mathcal{C}} \odot A_{22}$ are both causal attention matrices, and $A_{21}$ is an unmasked attention matrix.

---

**Algorithm 4:** CausalApproxD, recursive approximation of $D_{\mathcal{C}}$ for causal masking

1: **input**: matrices $Q, K \in \mathbb{R}^{n \times d}$
2: Split $Q$ and $K$ into equal sized sub-matrices: $Q = [Q_1^{\top}, Q_2^{\top}]^{\top}$ and $K = [K_1^{\top}, K_2^{\top}]^{\top}$
3: $\widetilde{D}_{\mathcal{C}11} \leftarrow$ CausalApproxD$(Q_1, K_1)$ and $\widetilde{D}_{\mathcal{C}22} \leftarrow$ CausalApproxD$(Q_2, K_2)$
4: Run the unmasked algorithm ApproxD (Algorithm 2) on $Q_2, K_1$ to get $\widetilde{D}_{21}$
5: **return** $\widetilde{D}_{\mathcal{C}} = \begin{bmatrix} \widetilde{D}_{\mathcal{C}11}, & \mathbf{0} \\ \mathbf{0}, & \widetilde{D}_{21} + \widetilde{D}_{\mathcal{C}22} \end{bmatrix}$

---

we can approximate its row sums using Algorithm 2. The two diagonal blocks $M_1^{\mathcal{C}} \odot A_{11}$ and $M_2^{\mathcal{C}} \odot A_{22}$ shown in Fig. 2 are causal attentions with half the original size. To handle these, we apply a recursive approach and further partition them into smaller blocks, and repeat this procedure. We present a pseudocode for this procedure in Algorithm 4.

## 4 EXPERIMENTS

In this section, we benchmark our algorithms by scaling up existing large language models to handle long-range sequences. All experiments are performed on a single A100 GPU with 40 GB memory and we use FlashAttention 2 (Dao, 2023) for the exact attention computation.

**Implementation Detail.** We implement HyperAttention based on sortLSH and uniform column sampling. Specifically, we first apply sortLSH to all rows in $Q, V \in \mathbb{R}^{n \times d}$. Then, each set of rows is partitioned into $b$ groups where $b$ is the block size as in Fig. 1. The $i$-th set of rows in $Q$ is multiplied by the corresponding set in $K$, resulting in a block-diagonal approximation of $A_{\mathcal{P}_Q, \mathcal{P}_K}$. Next, we optimize Algorithm 2 for approximating $D$ by sharing random indices $\{\ell_j\}_{j=1}^m$ with all rows in $Q$. This corresponds to uniformly sampling $m$ rows in $V$. To further simplify, we reuse indices $\{\ell_j\}_{j=1}^m$ for the Approximate Matrix Multiplication (AMM) in Lemma 2. The required operations involve permuting $n$ rows, reshaping tensors, and small matrix multiplications. Since every batch, head and block has the same configurations, the implementation can be parallelized using GPUs.

### 4.1 MONKEY PATCHING SELF-ATTENTION

We first evaluate HyperAttention on two pre-trained LLMs. We choose three models with different architectures that are widely used in practical applications: chatglm3-6b-32k (Du et al., 2022), and phi-1.5 (Li et al., 2023b). We patch their final $\ell$ attention layers by replacing with HyperAttentions where $\ell$ can vary from 0 to the number of all attention layers in each LLM. Note that attentions in both models requires causal masking and we make use of Algorithm 4 by recursively applying it until the input sequence lengths $n$ are less than 4,096. We set both bucket size $b$ and the number of sampled columns $m$ to 256 for all sequence lengths. We evaluate the performance of such monkey patched models in terms of perplexity and speedup.

We use LongBench (Bai et al., 2023), a collection of long context benchmark datasets, which contains 6 different tasks ranging from single and multiple-document question answering, summariza-

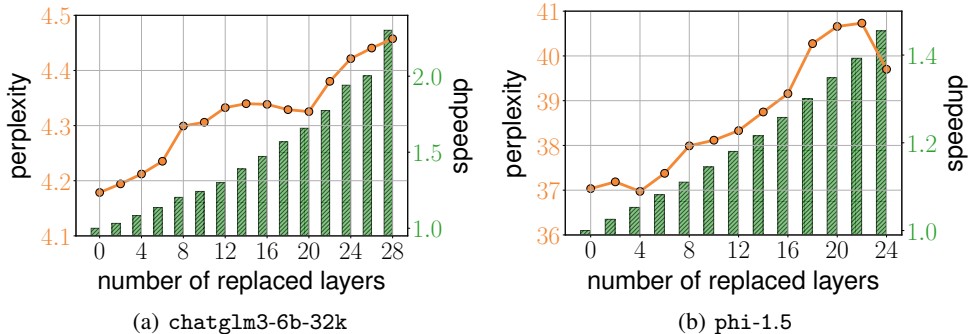

(a) `chatglm3-6b-32k`                                    (b) `phi-1.5`

Figure 3: Perplexity and speedup of `chatglm3-6b-32k` (left) and `phi-1.5` (right) monkey patched with HyperAttention. We vary the number of replaced attention layers in the final order.

| Number of | Task | | | | | |
|---|---|---|---|---|---|---|
| Replaced Layers | single-qa | multi-qa | summarization | few-shot | synthetic | code |
| 0 (exact) | 96.07 | 88.19 | 60.29 | 203.39 | 103.00 | 107.16 |
| 7 | 91.82 | 90.02 | 60.84 | 202.77 | 104.00 | 107.42 |
| 14 | 87.98 | 89.64 | 58.84 | 200.46 | 101.00 | 103.19 |
| 21 | 84.61 | 77.51 | 61.55 | 201.13 | 97.00 | 101.12 |
| 28 | 53.30 | 73.98 | 60.45 | 200.01 | 71.00 | 104.99 |

Table 1: Performance evaluation of `chatglm3-6b-32k` equipped with HyperAttentions on Long-Bench datasets (Bai et al., 2023). They contain 6 different tasks and we evaluate each of them with its own metric where higher value indicates better performance.

tion, few-shot learning, synthetic tasks, and code completion. We select a subset of dataset whose encoded sequence lengths are larger than 32,768 and trim them if the length is over 32,768 so that all data have sequence lengths of 32,768. Then, we compute the perplexity (i.e., loss on next tokens prediction) of each model. To highlight the scalability on the long sequences, we calculate the total speedup on all attention layers whether performed by HyperAttention or FlashAttention.

The results are summarized in Fig. 3. Observe that `chatglm3-6b-32k` shows a reasonable perplexity even after monkey patched by HyperAttention, e.g., after replacing 20 layers the perplexity increases approximately by 1 and it slowly goes up until 24 layers. But it improves runtimes in attention layers about 50%. If all the layers are replaced then the perplexity goes to up 12 but it runs about $2.3\times$ faster. For `phi-1.5`, similar happens but the perplexities are linearly increasing as the number of HyperAttention grows.

In addition, we evaluate the performances of monkey patched `chatglm3-6b-32k` on LongBench datasets and compute task-specific evaluation scores on each task including single-document question answering, multiple-document question answering, summarization, few-shot learning, synthetic tasks and code completion. Results are provided in Table 1. While replacing HyperAttention generally leads to performance degradation, we observe that its role can vary depending on the task at hand. For example, summarization and code completion are more robust to other tasks. Notably, when half of all attention layers are patched (i.e., 14 layers), we verify that most of the tasks do not degrade more than 13%. In particular, the performance of the summarization task remained almost unchanged, suggesting that this task may be more robust to partial alterations in the attention mechanism. We recall that computations in attention layers can be $1.5\times$ faster when $n = 32$k.

## 4.2 SINGLE SELF ATTENTION LAYER

We further explore the speedup of HyperAttention with varying sequence lengths from 4,096 to 131,072. We measure wall-clock times of both forward and forward+backward operations when they are computed with FlashAttention or are accelerated by HyperAttention. We measure the times with and without causal masking. All inputs $Q, K, V$ have the same length and their dimensions are fixed to $d = 64$ and the number of attention heads is set by 12. We chose the same parameters in

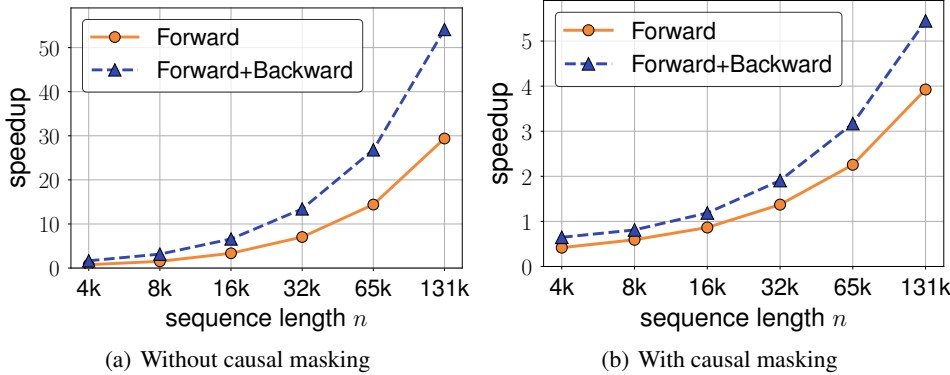

(a) Without causal masking         (b) With causal masking

Figure 4: Speedup of the exact computation using FlashAttention (Dao, 2023) and HyperAttention (this work) in single self-attention layer during forward and backward operations. For $n =$131k, HyperAttention runs up to $54\times$ faster without causal masking and $5.4\times$ with causal masking.

HyperAttention as described in the previous section. In Fig. 4, we observe that HyperAttention runs to up $54\times$ faster without causal masking and $5.4\times$ when the causal masking applies. Although time complexities of both causal masking and non-masking are the same, a practical algorithm for causal masking (Algorithm 1) requires additional operations such as partitioning $Q, K, V$, and merging attention outputs which result in an increase of practical runtime. However, those speedups will increase when the sequence length $n$ grows. We believe this opens the door to scale self-attention not only for inference but also for training or fine-tuning the LLMs to fit in significantly long sequences.

### 4.3 EMPIRICAL VERIFICATION OF ASSUMPTION

In addition, we empirically verify our assumption in Theorem 1, i.e., squared $\ell_2$-norm of columns in $D^{-1}A$ is upper bounded by $\frac{n^{o(1)}}{n}$. We investigate pretrained transformers with and without causal masking. For non-causal masking attention, we use the T2T-ViT model (Yuan et al., 2021) and take $Q, K, V$ from its first attention layer on the ImageNet test data set as it is the main computational bottleneck. For each image, we compute $\alpha$ to be the largest of the squared $\ell_2$-norms of the columns in $\left\| D^{-1}A \cdot e_i \right\|_2^2$ and collect the value over 50k images. The sequence length of the model is given by $n = 3{,}136$ and the averaged values of $\alpha$ is observed to be $8.1801$. This is much smaller than $n$ and can be possibly sublinear in $n$.

To further investigate the dependence on $n$, we utilize the `chatglm3-6b-32k` and LongBench narrative-qa dataset, changing the sequence length $n$ from 1k to 9k. We trim or pad the input context so that its length is strictly $n$. Unlike the vision model, we notice that the first columns in $D^{-1}A$ often contain heavy entries; hence we compute $\alpha$ as the largest squared norm excluding the first 32 columns. We collect these values for all heads and layers and compute their average. Fig. 5 plots the value of $\frac{\alpha}{n}$ with various sequence length $n$. It is observed that the value of $\frac{\alpha}{n}$ decreases as $n$ grows, supporting the claim that our assumption $\alpha = n^{o(1)}$ holds in practice.

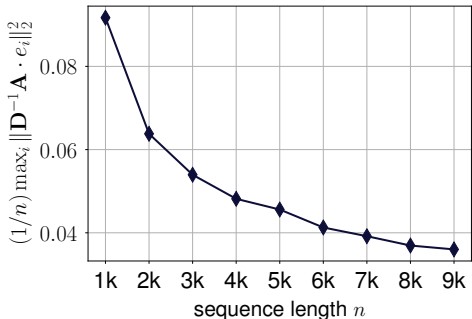

Figure 5: Empirical values of $\alpha$ (i.e., the largest squared $\ell_2$-norms of the columns in $D^{-1}A$). Theorem 1 assumes $\alpha = n^{o(1)}$ and the plot supports the claim that our assumption holds in practice.

## 5 CONCLUSION

In this work, we propose a simple linear time attention approximation algorithm by simplifying the existing algorithm based on kernel density estimation (KDE). We introduce a more general parameterization for a spectral approximation guarantee based on the condition number, which does not require assumptions used in prior work. Our algorithm makes use of sortLSH to find large entries and we adopt fast matrix multiplication via row norm sampling. We additionally study how our algorithm is used for causal masking by recursive partitioning. Empirically, we illustrate that pre-trained LLMs using our algorithm can enhance both inference and training speeds with only minimal performance degradation.

## ACKNOWLEDGEMENT

D. Woodruff worked on this while at Google Research and also while visiting the Simons Institute for the Theory of Computing.

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

## A  Omitted proofs

Here we include the proofs that were omitted in the main body of the paper. First, we present the proof of Lemma 1.

**Proof of Lemma 1:** First, we show that $\tau$ calculated in line 3 of Algorithm 2 is close to the maximum row sum of the matrix $(\mathbf{1}_n - \boldsymbol{M}^{\mathcal{H}}) \odot \boldsymbol{A}$. It is easy to check that $\frac{\tau}{\kappa} \leq \langle 1 - \boldsymbol{M}_{i,:}^{\mathcal{H}}, \exp(\boldsymbol{K}\boldsymbol{Q}_{i,:}^{\top}) \rangle \leq \tau\kappa$ for all $i \in [n]$ because of the definition of $\kappa$ in the lemma statement. Furthermore, if we define the set:

$$\mathbb{S}_0 := \left\{ i \in [n] : \langle 1 - \boldsymbol{M}_{i,:}^{\mathcal{H}}, \exp(\boldsymbol{K}\boldsymbol{Q}_{i,:}^{\top}) \rangle > \tau \right\}, \tag{4}$$

then we can show that $|\mathbb{S}_0|$ is small. Recall that $\tau$ is the maximum of the row sums of a random subset of rows, denoted by $\mathbb{T}$ where $|\mathbb{T}| = m$. Hence, $\Pr[|\mathbb{S}_0| \geq t] \leq (1 - t/n)^m$. Choosing $m = \Omega(\kappa^7 \alpha^2 \varepsilon^{-6} \log n)$ and $t = O(\kappa^{-7} \alpha^{-2} \varepsilon^6 n)$ gives that with probability at least $1 - \frac{1}{\text{poly}(n)}$

$$|\mathbb{S}_0| \leq O\left( \kappa^{-7} \cdot \alpha^{-2} \cdot \varepsilon^6 \cdot n \right). \tag{5}$$

Next, let us define the upper-capped version of matrix $\boldsymbol{A}$ where entries of $i$-th row on positions where the mask $\boldsymbol{M}^{\mathcal{H}}$ value is equal to zero are capped at value $C_i$ (line 6 of the algorithm) as:

$$\widetilde{\boldsymbol{A}} \in \mathbb{R}^{n \times n} : \ \widetilde{\boldsymbol{A}}_{i,j} := \begin{cases} \min\left(\boldsymbol{A}_{i,j}, C_i\right) & \text{if } \boldsymbol{M}_{i,j}^{\mathcal{H}} = 0 \\ \boldsymbol{A}_{i,j} & \text{otherwise} \end{cases}, \quad \text{for every } i, j \in [n].$$

We proceed by bounding the total mass of large entries of matrix $\boldsymbol{D}^{-1}\boldsymbol{A}$ lost through capping (i.e., entries of $\boldsymbol{A}$ that are larger than thresholds $C_i$). If we define constant $\widehat{C} := \frac{\varepsilon^2 m}{\kappa^2 n \log n}$, we can write,

$$
\begin{aligned}
\left\| \boldsymbol{D}^{-1}(\boldsymbol{A} - \widetilde{\boldsymbol{A}}) \right\|_1 &= \sum_{i,j \in [n]} (1 - \boldsymbol{M}_{i,j}^{\mathcal{H}}) \cdot (\boldsymbol{A}_{i,j} - \min(\boldsymbol{A}_{i,j}, C_i)) / \boldsymbol{D}_{i,i} \\
&= \sum_{t=0}^{\infty} \sum_{\substack{i,j \in [n] \\ (2^t - 1)\widehat{C}\boldsymbol{D}_{i,i} < \boldsymbol{A}_{i,j} - C_i \leq (2^{t+1} - 1)\widehat{C}\boldsymbol{D}_{i,i}}} \mathbf{1}_{\{\boldsymbol{M}_{i,j}^{\mathcal{H}} = 0\}} \cdot \frac{\boldsymbol{A}_{i,j} - \min(\boldsymbol{A}_{i,j}, C_i)}{\boldsymbol{D}_{i,i}} \\
&\leq \sum_{t=0}^{\log_2 \frac{\kappa^2}{\widehat{C}}} 2^{t+1}\widehat{C} \cdot \left| \left\{ i,j \in [n] : \boldsymbol{M}_{i,j}^{\mathcal{H}} = 0 \,\&\, \boldsymbol{A}_{i,j} > C_i + (2^t - 1)\widehat{C}\boldsymbol{D}_{i,i} \right\} \right| \\
&\leq \sum_{t=0}^{\log_2 \frac{\kappa^2}{\widehat{C}}} 2^{t+1}\widehat{C} \cdot \frac{\alpha}{(2^t \widehat{C})^2} = O\left( \frac{\varepsilon^4}{\kappa^5 \cdot \alpha} \cdot n \right)
\end{aligned}
\tag{6}
$$

The inequality in Eq. (6) follows because, for every $i \in [n]$, the cardinality of the set $\left\{ i,j \in [n] : \boldsymbol{M}_{i,j}^{\mathcal{H}} = 0 \,\&\, \boldsymbol{A}_{i,j} > C_i + (2^t - 1)\widehat{C}\boldsymbol{D}_{i,i} \right\}$ must be bounded by $\frac{\alpha}{(2^t \widehat{C})^2}$. The proof of this is by contradiction because otherwise, there must be an $l \in [n]$ such that the cardinality of the set $\mathbb{H}_l := \left\{ i \in [n] : \boldsymbol{M}_{i,l}^{\mathcal{H}} = 0 \bigwedge \boldsymbol{A}_{i,l} > C_i + (2^t - 1)\widehat{C}\boldsymbol{D}_{i,i} \right\}$ is at least $|\mathbb{H}_l| > \frac{\alpha}{n \cdot (2^t \widehat{C})^2}$. This implies that $\left\| \boldsymbol{D}^{-1}\boldsymbol{A} \cdot e^{(l)} \right\|_2^2 \geq \sum_{i \in \mathbb{H}_l} (\boldsymbol{A}_{i,l}/\boldsymbol{D}_{ii})^2 = \sum_{i \in \mathbb{H}_l} \left( \frac{C_i}{\boldsymbol{D}_{i,i}} + (2^t - 1)\widehat{C} \right)^2 \geq |\mathbb{H}_l| \cdot \left( \frac{\varepsilon^2 m}{\kappa^2 n \log n} + (2^t - 1)\widehat{C} \right)^2 > \frac{\alpha}{n}$, however, this contradicts with the precondition of the lemma about $\alpha = n \cdot \max_{i \in [n]} \left\| \boldsymbol{D}^{-1}\boldsymbol{A} \cdot e^{(i)} \right\|_2^2$. Now, if we defined the sets $\mathbb{S}_1, \mathbb{S}_2 \subseteq [n]$ as

$$
\mathbb{S}_1 = \left\{ i \in [n] : \frac{\varepsilon \boldsymbol{D}_{ii}}{3} < \|\boldsymbol{A}_{i,:} - \widetilde{\boldsymbol{A}}_{i,:}\|_1 \leq \frac{\boldsymbol{D}_{ii}}{3} \right\}, \quad \mathbb{S}_2 = \left\{ i \in [n] : \|\boldsymbol{A}_{i,:} - \widetilde{\boldsymbol{A}}_{i,:}\|_1 > \frac{\boldsymbol{D}_{ii}}{3} \right\}
\tag{7}
$$

then it follows from Eq. (6) that the cardinalities of $\mathbb{S}_1$ and $\mathbb{S}_2$ are bounded by

$$
|\mathbb{S}_1| \leq O\left( \kappa^{-4} \cdot \alpha^{-1} \cdot \varepsilon^3 n \right), \quad |\mathbb{S}_2| \leq O\left( \kappa^{-4} \cdot \alpha^{-1} \cdot \varepsilon^4 n \right).
\tag{8}
$$

Next note that $d_i$ computed in line 7 of the algorithm is an estimator for $i$-th row norm of the capped and masked matrix $(\mathbf{1}_n - \boldsymbol{M}^{\mathcal{H}}) \odot \widetilde{\boldsymbol{A}}$. Let us define an estimator for the unmasked capped matrix $\widetilde{\boldsymbol{A}}$ as $\widehat{d}_i := d_i + \langle \boldsymbol{M}_{i,:}^{\mathcal{H}}, \boldsymbol{A}_{i,:} \rangle$. By invoking Chernoff-Hoeffding inequality (see e.g., (McDiarmid, 1998)) along with union bound, because the lemma statement assumes that $m = \Omega\left( \frac{\kappa^7 \cdot \alpha^2}{\varepsilon^6} \log n \right)$ the following holds simultaneously for all $i \in [n] \setminus \mathbb{S}_2$ with probability $1 - \frac{1}{\text{poly}(n)}$:

$$
\frac{\left\| \widetilde{\boldsymbol{A}}_{i,:} \right\|_1}{1 + \varepsilon/6} \leq \widehat{d}_i \leq \frac{\left\| \widetilde{\boldsymbol{A}}_{i,:} \right\|_1}{1 - \varepsilon/6}.
\tag{9}
$$

This inequality combined with definition of $\mathbb{S}_1, \mathbb{S}_2$ in Eq. (7) implies that for any $i \in [n] \setminus (\mathbb{S}_1 \cup \mathbb{S}_2)$, $(1 - \varepsilon/2) \cdot \boldsymbol{D}_{i,i}^{-1} \leq \widehat{d}_i^{-1} \leq (1 + \varepsilon/2) \cdot \boldsymbol{D}_{i,i}^{-1}$. Now we bound the operator norm of the error as follows:

$$
\begin{aligned}
\left\| (\widetilde{\boldsymbol{D}}^{-1} - \boldsymbol{D}^{-1})\boldsymbol{A} \right\|_{\text{op}} &\leq \left\| \left( \widetilde{\boldsymbol{D}}^{-1} - \boldsymbol{D}^{-1} \right)_{\mathbb{S}_1 \cup \mathbb{S}_2} \boldsymbol{A} \right\|_{\text{op}} + \left\| \left( \widetilde{\boldsymbol{D}}^{-1} - \boldsymbol{D}^{-1} \right)_{[n] \setminus (\mathbb{S}_1 \cup \mathbb{S}_2)} \boldsymbol{A} \right\|_{\text{op}} \\
&\leq \frac{3}{2} \left\| \boldsymbol{D}_{\mathbb{S}_1}^{-1} \boldsymbol{A} \right\|_{\text{op}} + \left\| \left( \widetilde{\boldsymbol{D}}^{-1} - \boldsymbol{D}^{-1} \right)_{\mathbb{S}_2} \boldsymbol{A} \right\|_{\text{op}} + \frac{\varepsilon}{2} \left\| \boldsymbol{D}_{[n] \setminus (\mathbb{S}_1 \cup \mathbb{S}_2)}^{-1} \boldsymbol{A} \right\|_{\text{op}} \\
&\leq \frac{3}{2} \left\| \boldsymbol{D}_{\mathbb{S}_1}^{-1} \boldsymbol{A} \right\|_{\text{op}} + \kappa^2 \left\| \boldsymbol{D}_{\mathbb{S}_2 \cap \mathbb{S}_0}^{-1} \boldsymbol{A} \right\|_{\text{op}} + \kappa \left\| \boldsymbol{D}_{\mathbb{S}_2 \setminus \mathbb{S}_0}^{-1} \boldsymbol{A} \right\|_{\text{op}} + \frac{\varepsilon}{2} \left\| \boldsymbol{D}_{[n] \setminus (\mathbb{S}_1 \cup \mathbb{S}_2)}^{-1} \boldsymbol{A} \right\|_{\text{op}} \\
&\leq \frac{3}{2} \left\| \boldsymbol{D}_{\mathbb{S}_1}^{-1} \boldsymbol{A} \right\|_{\text{op}} + \kappa^2 \left\| \boldsymbol{D}_{\mathbb{S}_0}^{-1} \boldsymbol{A} \right\|_{\text{op}} + \kappa \left\| \boldsymbol{D}_{\mathbb{S}_2}^{-1} \boldsymbol{A} \right\|_{\text{op}} + \frac{\varepsilon}{2} \left\| \boldsymbol{D}_{[n] \setminus (\mathbb{S}_1 \cup \mathbb{S}_2)}^{-1} \boldsymbol{A} \right\|_{\text{op}},
\end{aligned}
\tag{10}
$$

where the second inequality above follows from the inequality in Eq. (9) and also because the definition of $\mathbb{S}_1$ ensures that $\widetilde{D}_{i,i} \geq (1 - 1/3) \cdot D_{i,i}$ for any $i \in \mathbb{S}_1$. The third inequality above follows because the lower capping in line 8 of the algorithm ensures that $\widetilde{D}_{j,j} \geq \langle M_{j,:}^{\mathcal{H}}, A_{j,:} \rangle + \tau/\kappa$ for any $j \in \mathbb{S}_2$ while $D_{j,j} \leq \langle M_{j,:}^{\mathcal{H}}, A_{j,:} \rangle + \tau$ for any $j \notin \mathbb{S}_0$ by definition of set $\mathbb{S}_0$ and we know that $D_{j,j} \leq \langle M_{j,:}^{\mathcal{H}}, A_{j,:} \rangle + \tau\kappa$ for $j \in \mathbb{S}_0$.

Finally we conclude the proof by bounding the terms $\left\| D_{\mathbb{S}_r}^{-1} A \right\|_{\mathrm{op}}$ for $r \in \{0, 1, 2\}$ in Eq. (10). Fix some $r \in \{0, 1, 2\}$. Let $v$ be the unit-normed vector that realizes the operator norm of $D_{\mathbb{S}_r}^{-1} A$. Since $D_{\mathbb{S}_r}^{-1} A$ is a non-negative matrix, w.l.o.g., we can assume that $v$ is a non-negative vector. More precisely $v := \arg\max_{\substack{x \in \mathbb{R}_+^n \\ \|x\|_2 = 1}} \left\| D_{\mathbb{S}_r}^{-1} A \cdot x \right\|_2$. One has that $\left\| D_{\mathbb{S}_r}^{-1} A \cdot v \right\|_2 = \left\| D_{\mathbb{S}_r}^{-1} A \right\|_{\mathrm{op}}$. We define the sequence of binary matrices $B^0, B^1, B^2 \ldots$ which have same shape as $D_{\mathbb{S}_r}^{-1} A$ as follows:

$$B_{i,j}^t := \mathbf{1}_{\left\{ 2^{-t-1}\sqrt{\alpha/n} < \left[ D_{\mathbb{S}_r}^{-1} A \right]_{i,j} \leq 2^{-t}\sqrt{\alpha/n} \right\}} \qquad \text{for every integers } t \geq 0. \tag{11}$$

Note that because of the precondition of the lemma about $\alpha = n \cdot \max_{i \in [n]} \left\| D^{-1} A \cdot e^{(i)} \right\|_2^2$ which implies $\left[ D^{-1} A \right]_{i,j} \leq \sqrt{\alpha/n}$, we have the following inequality on each entry of the matrix $D_{\mathbb{S}_r}^{-1} A$:

$$\left[ D_{\mathbb{S}_r}^{-1} A \right]_{i,j} \leq \sqrt{\alpha/n} \cdot \sum_{t=0}^{\infty} 2^{-t} \cdot \left[ B^t \right]_{i,j}.$$

Since $D_{\mathbb{S}_r}^{-1} A$ and $v$ both have non-negative entries, the above inequality implies the following:

$$\left\| D_{\mathbb{S}_r}^{-1} A \cdot v \right\|_2 \leq \sqrt{\alpha/n} \cdot \left\| \sum_{t=0}^{\infty} 2^{-t} \cdot B^t \cdot v \right\|_2 \leq \sqrt{\alpha/n} \cdot \sum_{t=0}^{\infty} 2^{-t} \cdot \left\| B^t \cdot v \right\|_2. \tag{12}$$

Now to bound $\left\| B^t \cdot v \right\|_2$ we first find bounds on the number of 1's in rows and columns of $B^t$. Using the definition of $B^t$ in Eq. (11) and the fact that row sums in matrix $D^{-1} A$ are equal to 1, we have:

$$\left\| B_{i,:}^t \right\|_0 \leq \min(2^{t+1}\sqrt{n/\alpha}, n). \tag{13}$$

Additionally, using the precondition of the lemma about $\alpha = n \cdot \max_{i \in [n]} \left\| D^{-1} A \cdot e^{(i)} \right\|_2^2$, we have:

$$\left\| B_{:,j}^t \right\|_0 \leq \min(2^{2t+2}, |\mathbb{S}_r|). \tag{14}$$

Now we bound the norm $\left\| B^t \cdot v \right\|_2$ for an arbitrary integer $t \geq 0$ as follows:

$$
\begin{aligned}
\left\| B^t \cdot v \right\|_2^2 &\leq \sum_{i=1}^{|\mathbb{S}_r|} \left\| B_{i,:}^t \right\|_0 \cdot \left\| B_{i,:}^t \odot v \right\|_2^2 && \text{(Cauchy–Schwarz inequality)} \\
&\leq 2^{t+1}\sqrt{n/\alpha} \cdot \sum_{i=1}^{|\mathbb{S}_r|} \left\| B_{i,:}^t \odot v \right\|_2^2 = 2^{t+1}\sqrt{n/\alpha} \cdot \sum_{j \in [n]} \sum_{i=1}^{|\mathbb{S}_r|} B_{i,j}^t \cdot v_j^2 \\
&= 2^{t+1}\sqrt{n/\alpha} \cdot \sum_{j \in [n]} \left\| B_{:,j}^t \right\|_0 \cdot v_j^2 \\
&\leq 2^{t+1}\sqrt{n/\alpha} \cdot \min(2^{2t+2}, |\mathbb{S}_r|) \cdot \sum_{j \in [n]} v_j^2 = 2^{t+1}\sqrt{n/\alpha} \cdot \min(2^{2t+2}, |\mathbb{S}_r|),
\end{aligned}
$$

where the inequality in second line above follows from Eq. (13) and the inequality in the last line follows from Eq. (14). The last equality follows from the assumption that $\|v\|_2 = 1$. Therefore,

$$\left\| B^t \cdot v \right\|_2 \leq \begin{cases} 2^{\frac{3t+3}{2}} \cdot (n/\alpha)^{1/4} & \text{if } 2^{t+1} \leq \sqrt{|\mathbb{S}_r|} \\ 2^{\frac{t+1}{2}} \cdot (n/\alpha)^{1/4} \cdot \sqrt{|\mathbb{S}_r|} & \text{otherwise} \end{cases}.$$

Now by plugging the above inequalities into Eq. (12) we find that:

$$\left\| \boldsymbol{D}_{\mathbb{S}_r}^{-1} \boldsymbol{A} \right\|_{\text{op}} \leq \sqrt{\alpha/n} \cdot \left( \sum_{t=0}^{\log_2 \sqrt{|\mathbb{S}_r|}-1} 2^{-t} \cdot \left\| B^t \cdot v \right\|_2 + \sum_{t=\log_2 \sqrt{|\mathbb{S}_r|}}^{\infty} 2^{-t} \cdot \left\| B^t \cdot v \right\|_2 \right)$$

$$\leq 12 \left( \frac{\alpha \cdot |\mathbb{S}_r|}{n} \right)^{1/4} \leq \begin{cases} \frac{\varepsilon^{3/2}}{6\kappa^{7/4}\alpha^{1/4}} & \text{if } r = 0 \\ \frac{\varepsilon^{3/4}}{9\kappa} & \text{if } r = 1 \\ \frac{\varepsilon}{6\kappa} & \text{if } r = 2 \end{cases}$$

where the last line above follows from the upper bound on the size of sets $\mathbb{S}_r$ we obtained in Eq. (5) and Eq. (8). Finally, by plugging the above inequality into Eq. (10) and using the fact that $\boldsymbol{D}^{-1}\boldsymbol{A}$ is a row-stochastic matrix and thus $\left\| \boldsymbol{D}^{-1}\boldsymbol{A} \right\|_{\text{op}} \geq 1$ the lemma follows.

$\square$

Next, we prove the main theorem.

**Proof of Theorem 1:** The diagonal matrix $\widetilde{\boldsymbol{D}}$ in line 2 is computed by invoking Algorithm 2. By Lemma 1 with probability at least $1 - \frac{1}{\text{poly}(n)}$, it holds that $\left\| \left( \widetilde{\boldsymbol{D}}^{-1} - \boldsymbol{D}^{-1} \right) \boldsymbol{A} \right\|_{\text{op}} \leq \varepsilon/2 \cdot \left\| \boldsymbol{D}^{-1}\boldsymbol{A} \right\|_{\text{op}}$. Furthermore, in line 3 of the algorithm $\boldsymbol{S}$ is defined as the sampling matrix according to the row norms of $\boldsymbol{V}$. To invoke Lemma 2, we need to have a bound on the stable rank of $\widetilde{\boldsymbol{D}}^{-1}\boldsymbol{A}$. First, from Lemma 1 we know that $\left\| \widetilde{\boldsymbol{D}}^{-1}\boldsymbol{A} \right\|_{\text{op}} \geq (1-\varepsilon/2) \cdot \left\| \boldsymbol{D}^{-1}\boldsymbol{A} \right\|_{\text{op}} \geq 1/2$, where the second inequality follows because $\boldsymbol{D}^{-1}\boldsymbol{A}$ is a row-stochastic matrix. Therefore we have $\text{srank}(\widetilde{\boldsymbol{D}}^{-1}\boldsymbol{A}) \leq 4\left\| \widetilde{\boldsymbol{D}}^{-1}\boldsymbol{A} \right\|_F^2$. Second, the lower capping in line 8 of Algorithm 2 ensures that $\left\| \widetilde{\boldsymbol{D}}^{-1}\boldsymbol{A} \right\|_F^2 \leq \kappa^2 \left\| \boldsymbol{D}^{-1}\boldsymbol{A} \right\|_F^2 \leq n^{o(1)} \cdot \left\| \boldsymbol{D}^{-1}\boldsymbol{A} \right\|_F^2 \leq n^{o(1)}$, thus $\text{srank}(\widetilde{\boldsymbol{D}}^{-1}\boldsymbol{A}) \leq n^{o(1)}$.

With this bound on the stable rank and since $m = d \cdot n^{o(1)} = \Omega\left( \varepsilon^{-2} d \cdot \text{srank}(\widetilde{\boldsymbol{D}}^{-1}\boldsymbol{A}) \right)$, Lemma 2 implies that $\left\| \widetilde{\boldsymbol{D}}^{-1}\boldsymbol{A}\boldsymbol{S}^{\top} \cdot \boldsymbol{S}\boldsymbol{V} - \widetilde{\boldsymbol{D}}^{-1}\boldsymbol{A}\boldsymbol{V} \right\|_{\text{op}} \leq \varepsilon/3 \cdot \left\| \widetilde{\boldsymbol{D}}^{-1}\boldsymbol{A} \right\|_{\text{op}} \|\boldsymbol{V}\|_{\text{op}} \leq \varepsilon/2 \cdot \left\| \boldsymbol{D}^{-1}\boldsymbol{A} \right\|_{\text{op}} \|\boldsymbol{V}\|_{\text{op}}$ with probability 0.99.

By combining these two inequalities, we obtain the spectral approximation guarantee in Eq. (1). The runtime of this algorithm is primarily determined by the time it takes to invoke Algorithm 2 in line 2, which is dominated by the time to multiply two matrices of sizes $n \times d$ and $d \times m$ and the time to calculate attention matrix entries at the positions defined by $\boldsymbol{M}^{\mathcal{H}}$. Using the matrix multiplication result from (Le Gall, 2012), the first computation can be done in time $O(dn^{1+o(1)})$ and the latter can be done in $\text{nnz}(\boldsymbol{M}^{\mathcal{H}})$. $\square$

**Proof of Corollary 1:** Because $r = \log_2 n$, we have $\Pr[\mathcal{H}(\boldsymbol{Q}_{i,*}) = \mathcal{H}(\boldsymbol{K}_{j,*})] \leq 1/n^{1-o(1)}$ whenever $\theta(\boldsymbol{Q}_{i,*}, \boldsymbol{K}_{j,*}) \geq \frac{\pi}{2}(1 - o(1))$. As there are at most $n^2$ total pairs, the expected number of such pairs that collide under $\mathcal{H}$ is at most $n^{1+o(1)}$ and so by a Markov bound is at most $n^{1+o(1)}$ with failure probability $1/n^{o(1)}$.

Since we also assume there are at most $n^{1+o(1)}$ pairs $(i,j)$ with $\theta(\boldsymbol{Q}_{i,*}, \boldsymbol{K}_{j,*}) < \frac{\pi}{2}(1 - o(1))$, there can be at most $n^{1+o(1)}$ additional pairs that collide.

Thus, in total we have $n^{1+o(1)}$ collisions, and consequently the number of non-zero entries in $\boldsymbol{M}^{\mathcal{H}}$ is at most $n^{1+o(1)}$ with failure probability $1/n^{o(1)}$. The proof now follows by the assumptions of the corollary statement as well as Theorem 1. $\square$

**Proof of Corollary 2:** Let $\boldsymbol{T}$ be an ExpanderSketch matrix with $O(\tau \log n)$ rows. By the guarantees (Larsen et al., 2016) of $\boldsymbol{T}$, we have that with probability at least $1 - \frac{1}{n^3}$, for any fixed vector $x \in \mathbb{R}^n$, that from $\boldsymbol{T} \cdot x$, one can recover a set $S$ of indices $i \in [n]$ such that if $x_i^2 \geq \frac{\|x\|_2^2}{\tau}$, then $i \in S$, whereas if $x_i^2 \leq \frac{\|x\|_2^2}{2\tau}$, then $i \notin S$. Further, $S$ can be found from $\boldsymbol{T}x$ in $n^{o(1)}$ time.

We compute $\boldsymbol{T} \cdot \boldsymbol{Q}$, followed by $(\boldsymbol{T} \cdot \boldsymbol{Q}) \cdot \boldsymbol{K}^\top$. Note that the time for this computation is $O(\tau n \log n)$. Next, for each column $j$ of $\boldsymbol{Q} \boldsymbol{K}^\top$ this allows us to construct a set $S_j$ with the property that if $(\boldsymbol{Q} \cdot \boldsymbol{K}^\top)_{i,j} \geq \frac{\|\boldsymbol{Q} \cdot \boldsymbol{K}^\top e_j\|_2^2}{\tau}$, then $i \in S_j$. This holds simultaneously for all columns with probability at least $1 - \frac{1}{n^2}$ by a union bound. The time for constructing all the sets $S_j$ is $n^{1+o(1)}d$

Note that $|S_j| \leq 2\tau$ for all $j$, and we can explicitly compute the exact value of $(\boldsymbol{Q} \cdot \boldsymbol{K}^\top)_{i,j}$ for all $i \in S_j$ and all $j$, in $O(n\tau d)$ time. By the assumptions of the corollary, we have that $S_j$ contains a superset of the support of the $j$-th column of $\boldsymbol{M}^{\mathcal{H}}$, and since we can compute the values exactly, we can exactly construct the mask $\boldsymbol{M}^{\mathcal{H}}$ matrix that the corollary requires, and in $n^{1+o(1)}d$ time. The proof now follows by the assumptions of the statement as well as Theorem 1. □

