# OpenReview forum: "HyperAttention: Long-context Attention in Near-Linear Time"
_ICLR.cc/2024/Conference — ICLR 2024 poster_

### Official Review · Reviewer_xXnD · 2023-10-31

**Soundness:** 3 good
**Presentation:** 3 good
**Contribution:** 3 good
**Rating:** 5
**Confidence:** 4

**Summary:**

The authors propose a method to reduce the computation complexity of transformers down from quadratic in squence length to near liner. The proposed idea proposes to perform sampling to approximate Attention. The authors claim that compared to prior works there proposed idea is practical and is able to outperform existing approaches.

The core idea is to perform sampling of the value matrix based on the norms and the \sigma(QK^T) matrix to come up with important entries.

**Strengths:**

1. I think this a resonably effective idea. The bounds provided are kind of standard and the paper is well structured.
2. The idea of separating into multiple matrix when performing causal masking is very interesting. I think this is a very nice contribution.

**Weaknesses:**

Before highlighting the weaknesses I would like to point out where my criticism's are coming from. More often then not in Machine Learning approximation results, I have personally observed that the approximation often takes more time than the actual operation, often due to hardware constraints, which makes the approach moot and impractical. Stating this most my questions are coming from a point of view to discern how realistic will these speedups be in real world. I hope the authors are not too bothered by my naive questions.

1. My first question is regarding evaluation. Are the speedups proposed end to end. As in the runtime reported in Figure 3(a) and 3(b) Figure 4 include the run time for generating the mask M_h . The reason I am a bit skeptical is that approximation using Algoirthm 1 does not look very GPU friendly to my naive understanding. And you have mentioned in your contributions (On Page 3) "We assume these procedures are fast, and that after removing the heavy entries, two parameters in the resulting attention matrix are small: (1) the max column l1-norm, and (2) the ratio of row norms in the un-normalized attention matrix." This makes me wonder if you include the runtime. I would ideally love to see a breakdown of the time spent in different procedures as that will help new and unfamiliar readers to understand how with your novel approach bottlenecks have shifted and where the new bottlenecks are.

2. For your proofs you assume the following - "To be more precise, we assume that for any i ∈ [n] there exists some α = no(1) such that D−1A · e(i) 2 ≤ αn ." Can you verify this assumption experimentally. I have generally found a proof to stand the test of time a bit more if the assumptions have some sort of experimental verification.

3. Do the authors have some understanding if there is better way of choosing the number of layers to use for their proposed "HyperAttention" mechansim. It currently feels arbitary to choose these layers. It would be interesting to understanding why it works in certain cases and where it doesn't.

4. The other thing I found missing is the "m" value (Algorithm 2). Can authors enumerate the "m" values used in their setup. Is it based on the bound or is it another hyper-parameter. Especially for long sequence results, since the accuracy can not be verified there it would be interesting to see the parameters which authors have used.


In general I found the paper easy to follow and appreciate the various figure the authors have included. I like the experiments which authors have performed and showed how on long sequence length their method is more useful. I am happy to bump up the score to a weak accept or accept based on authors reponses. Nice work !!.

**Questions:**

See weakness section.

---

> ### Author Response · Authors · 2023-11-20
> **Official Comment by Authors**
>
> > 1. My first question is regarding evaluation. ... and where the new bottlenecks are.
>
> - All runtimes in our results are end-to-end. Although in the right-most figure in Figure 1 the sparsified attention by ${\mathbf {M}}^{\mathcal{H}}$ seems to be computationally inefficient and not parallelizable, we follow the operations in the middle figure in practice, which does not compute the mask ${\mathbf  M}^{\mathcal{H}}$ explicitly. More specifically, rows in $\mathbf{Q}$ and $\mathbf{K}$ are sorted using Hamming LSH, then they are partitioned into “$b$” (i.e., the block size in Figure 1) groups.
> Finally,  the i-th set of rows in $\mathbf{Q}$ is multiplied by the corresponding set in $\mathbf{K}$, resulting in a block-diagonal approximation of $\mathbf{A}_{P_Q, P_K}$ . The required operations involve permuting $n$ rows, reshaping tensors, and small matrix multiplications. Since every batch, head and block has the same configurations, the implementation can be parallelized using GPUs. We have added more details in the updated submission.
>
> - The two parameters $\alpha$ and $\kappa$ are used to analyze the runtime of our algorithm but they are not required to compute in our practical algorithm.

---

> ### Author Response · Authors · 2023-11-20
> **Official Comment by Authors**
>
> > 2. For your proofs you assume the following  ... the assumptions have some sort of experimental verification.
>
> - We additionally conducted experiments to verify our assumption. For the attention with non-causal masking, we use the T2T-ViT model and take query, key and values from its first attention layer on the ImageNet test data set. Then, we compute $\alpha$ to be the maximum of the squared $\ell_2$-norms of the columns in $\mathbf{D}^{-1} \mathbf{A}$. The sequence length of T2T-ViT is $n=3136$ and the average value of $\alpha$ is observed to be 8.1801, which can be sublinear in $n$.
>
> - To further investigate the dependence on $n$, we choose the ChatGLM and longbench narrative-qa dataset used in the submission and change the sequence length n from 1k to 9k. As a result, we observe that the value of $\alpha$ increases sub-linearly in $n$, as below. This supports the claim that our assumption holds in practice. We have added these results in our updated submission.
>
>   | $n$  |$\alpha / n$|
>   | :------------- |:-------------:|
>   | 1024      |  0.09170  |
>   | 2048      |  0.06379  |
>   | 3072      |  0.05395  |
>   | 4096      |  0.04817  |
>   | 5120      |  0.04559  |
>   | 6144      |  0.04128  |
>   | 7168      |  0.03918  |
>   | 8192      |  0.03694  |
>   | 9216      |  0.03601  |

---

> ### Author Response · Authors · 2023-11-20
> **Official Comment by Authors**
>
> > 3. Do the authors have some understanding ... It would be interesting to understanding why it works in certain cases and where it doesn't.
>
> - In our experiments, we choose to replace the final L-layers where L changes from 1 to the total number of layers in the given pretrained Transformers, e.g.,  [1]. We motivate this setting from fine-tuning methods where the last few layers are updated. We believe that finding better configurations of layers to approximate is a very interesting future problem.
>
>   [1] Chen, Boqi, Fandi Yi, and Dániel Varró. "Prompting or Fine-tuning? A Comparative Study of Large Language Models for Taxonomy Construction." https://arxiv.org/pdf/2309.01715.pdf.

---

> ### Author Response · Authors · 2023-11-20
>
> > 4. The other thing I found missing is the "m" value (Algorithm 2). Can authors enumerate the "m" values used in their setup. Is it based on the bound or is it another hyper-parameter. Especially for long sequence results, since the accuracy can not be verified there it would be interesting to see the parameters which authors have used.
>
> - In our experiments, we consider the value “$m$” to be a hyper-parameter, which corresponds to the number of columns in $\mathbf{D}^{-1} \mathbf{A}$ selected uniformly at random. Motivating our analysis that m can be sublinear in “$n$”, we fix this value to be 256 for all the experiments. We observe that a larger value of m shows a better performance on LLMs (e.g., perplexity) but it makes our algorithm slower.

---

> ### Author Response · Authors · 2023-11-22
> **Discussion ends in 2 days**
>
> Dear reviewer,
>
> Since you indicated that you were
>
> > happy to bump up the score to a weak accept or accept based on authors reponses.
>
> we wanted to inquire whether you were satisfied with our responses. If not, please let us know soon so that we could provide more details or address your other questions/concerns before the discussion period ends soon.
>
> Thank you again for your time!

---

> > ### Comment · Reviewer_xXnD · 2023-12-03
> > **Thank you.**
> >
> > Note for AC.
> >
> > (i) The authors have clarified that their time is end to end. However, the authors response is not particularly inspiring as they talk about how a version of algorithm is amenable. "Since every batch, head and block has the same configurations, the implementation can be parallelized using GPUs. ". They have not provided breakdowns. I am feeling a little unsure of accepting this paper without it.
> >
> >
> > (ii) I appreciate the response for experimental verification of proof.
> >
> >
> > (iii) My current concerns are mainly about hyper-parameter selection and time breakdowns.
> >
> > Thank you

---

### Official Review · Reviewer_NFiG · 2023-11-02

**Soundness:** 3 good
**Presentation:** 3 good
**Contribution:** 3 good
**Rating:** 8
**Confidence:** 3

**Summary:**

Authors propose a new LSH-based approximation algorithm for attention layer, called HyperAttention. Similar to KDEformer, authors aim to bound the operator norm of the error, and this is done by two-steps: approximating the normalization matrix, and the attention matrix. Both approximations are primarily based on sortLSH, as with KDEformer, although authors generalize the analysis to be applicable to other sketching algorithms. Subquadratic runtime is proved. For empirical experiments, authors evaluate with LongBench tasks, perplexity on LongBench. Wall-clock time of HyperAttention on 131k sequence is also measured.

**Strengths:**

Significance: This paper provides a new theoretically-grounded approximation algorithm for attention layers. The algorithm doesn't rely on kernel density estimation, and the analysis is more straightforward to associate the assumed properties of attention matrix with theoretical guarantees. Hence, future work will find it useful to build upon this work to adapt assumptions and derive new results.

Originality: Although the algorithm is based on the same primitive sortLSH, as prior work KDEformer did, the actual algorithm is new. Also, authors generalize to other sketching algorithms as well, which makes a stronger connection to sketching literature.

Clarity: The overall proof strategy is clearly described. High-level intuition and interpretation of proof parameters are explained such that readers shall follow the main logic without having to read the proof. However, intuitive explanation of algorithms is mostly reserved to appendix, which is unfortunate but understandable given page constraints.

Quality: I wasn't able to closely verify proofs, but overall proof strategy of the work seems sound.

**Weaknesses:**

Empirical evaluations of this paper is not aligned well with prior work, which makes it difficult for readers to understand the practical usefulness of the proposed approach. For example, prior work KDEformer evaluates on BigGAN, ImageNet Classification, and Long Range Arena Benchmark, comparing against not only exact baseline but stronger baselines such as Reformer/Performer. While HyperAttention shared many similarity with KDEformer and claims to be more practical due to not using KDE, HyperAttention is not compared against KDEformer, making it difficult to see whether the difference between to actually make a practical difference. Also, end-to-end training experiment is lacking, and only forward/backward computation time was evaluated.

Although HyperAttention shares the overall proof strategy with KDEformer, they differ significantly in assumptions they make. I believe it is potentially a strength, as HyperAttention shall make more realistic assumptions and strengthen guarantees. However, it wasn't clear how HyperAttention's assumptions relate to KDEformer. A more detailed discussion of comparison between KDEformer and HyperAttention will clarify the theoretical contribution of this paper despite sharing the overall proof strategy.

**Questions:**

Eq (10) in Section 1.2 probably mean Eq(1).

In equation (5) in Appendix A, which result are authors using to prove this?

---

> ### Author Response · Authors · 2023-11-20
> **Official Comment by Authors**
>
> > Empirical evaluations of this paper is not aligned well with prior work, .... A more detailed discussion of comparison between KDEformer and HyperAttention will clarify the theoretical contribution of this paper despite sharing the overall proof strategy.
>
> - We thank the reviewer for their insightful comments. As the reviewer points out, HyperAttention is much simpler and more streamlined than the KDEformer. Particularly, the KDEformer requires a kernel density estimation (KDE) algorithm which is impractical and fairly hard to implement in general, in particular, with GPUs. We streamline such a complicated KDE to a simple approximation via sortLSH and uniform sampling. The assumptions we use are for guaranteeing that our uniform sampling approach can well-approximate the kernel density. We empirically observe that these assumptions hold in practice (please refer to our responses to reviewer **xXnD**).
>
>
> - Additionally, KDEformer does not support **causal masking** and this might be the reason why it mostly works on vision transformers, which does not require causal masking. In this work, we propose a novel method (HyperAttention) to support causal masking, and it allows us to apply it to causal language models. Furthermore, HyperAttention can be implemented in a modular way,  thereby allowing us to utilize other accelerated self-attention algorithms such as FlashAttention. This makes the gap in runtimes between HyperAttention and KDEformer even larger.
>
> - To see the practical differences between HyperAttention and KDEformer, we conduct ImageNet classification experiments with the T2T-ViT model. We apply HyperAttention and KDEformer to the first attention layer because it is the main computational bottleneck. The sequence length is 3136 and we set all hyperparameters in both methods so that they have the same memory and time complexity. The per-image runtime of KDEformer in the attention layer was 13.175 ms while HyperAttention shows 8.634 ms runtime (x1.53 faster). The test accuracies of KDEformer and HyperAttention were 81.11% and 81.14%, respectively, which are the same within a standard deviation.
>
> > Eq (10) in Section 1.2 probably mean Eq(1).
>
> - We appreciate your bringing attention to this typo. Equation (1) is indeed correct and we have corrected it in our updated submission.
>
> > In equation (5) in Appendix A, which result are authors using to prove this?
>
> - Equation (5) shows an upper bound on the number of rows for which the row sum is greater than $\tau$. To derive this bound, we make use of the fact that tau is the maximum of the row sums of a random subset of rows, denoted by T. Recall that $|T|=m$. Now note that the probability that the size of the set $S_0$ is greater than $t$ for some $t$, equals $\Pr[|S_0| \ge t] \le (1-t/n)^{m}$. Plugging $m=\Omega(\kappa^7 \alpha^2 \varepsilon^{-6} \log n)$ and $t=O(\kappa^{-7} \alpha^{-2} \varepsilon^6 n)$ gives Equation (5) with probability $1-1/\text{poly}(n)$. We have updated this in our submission file.

---

> > ### Comment · Reviewer_NFiG · 2023-12-04
> >
> > Thanks for addressing my concerns. These explanations make sense to me, and I hope other readers find them helpful as well. I will increase my score accordingly.

---

### Official Review · Reviewer_cCwA · 2023-11-04

**Soundness:** 3 good
**Presentation:** 2 fair
**Contribution:** 3 good
**Rating:** 6
**Confidence:** 3

**Summary:**

The paper proposes a new approach of accelerating the computation of attention layer. The proposed method has near linear time (under plausible assumption on column norm) and demonstrates significant advantage in experiments.

The attention layer in Transformer requires $\Omega(n^2)$ computation given input sequence of length $n$, this costs is prohibitive for long sequence and there are many recent work on reducing this computation cost. The paper proposes a new approach that uses near linear time, when the column norm of the attention matrix (i.e., $\exp(KQ)$, where $K$ is the key matrix and $Q$ is the query matrix at attention layer) is small (or balance). This is a challenging task as one needs to take account into the softmax function. The algorithm first uses LSH to identify large entries, then apply fast approximate matrix multiplication by random sampling rows.

The paper also performs empirical study on real word dataset. When the sequence length is roungly 30k, the inference time decreases roughly 50% with a slight increase of perplexity (roughly 20% -- 50%, depend on the dataset and the number of layer replaced). The speedup is significant for longer sequence with one single transformer layer (up to 50x without causual masking and 5x with causual masking).

--------------------------------------------------------------------
I have read the author's response and I would keep my positive evaluation towards the paper. Personally I feel the paper could benefit from conducting more extensive experiments, but I feel the idea itself sounds interesting and it is worth publication.

**Strengths:**

The proposed approach has nearly linear runtime in theory (ableit with some assumptions) and has good performance in practice.

**Weaknesses:**

There is no major weakness.

The presentation is good overall, but the theory part could be improved.
For example, add in-line comments for algorithm description and give explanation on which steps the assumption are required.

**Questions:**

.

---

> ### Author Response · Authors · 2023-11-20
> **Official Comment by Authors**
>
> We thank the reviewer for their detailed and insightful comments. We have modified the submission to include in-line comments describing the purpose of each quantity in algorithm 2, as well as an explanation before the algorithm of where the assumptions of Lemma 1 are used; in particular, the assumptions are used in the proofs to show that the variance of the estimator is small, therefore allowing for small sample complexity.

---

### Public Comment · ~Oliver_Bennett1 · 2023-11-13
**Public Comment**

Dear authors,

[1] as well proposed to utilize the sampling techniques to approximate softmax attention from a sketching perspective, and accordingly provide approximation error guarantees. This very related reference could be considered for clarifying the difference and the novelty.

Thanks.

[1] Sketching as a Tool for Understanding and Accelerating Self-attention for Long Sequences. NAACL 2022.

---

> ### Author Response · Authors · 2023-11-20
> **Thanks**
>
> Thanks for sharing this work. Here we give a comparison to the paper [1]
>
> - That paper requires not only estimating the row norms of $\mathbf{V}$, but also estimating the column norms of $\mathbf{A}$, where $\mathbf{A} = \exp(\mathbf{QK}^T)$. One of our key observations is that for the approximate matrix product guarantee, i.e., for preserving $\mathbf{D}^{-1} \mathbf{A SS^T V}$ up to the same desired additive error as in [1], one needs only to sample from the row norms of $\mathbf{V}$ and *does not need column norm information* of $\mathbf{D^{-1} A}$ (we note that the failure probability dependence is worse, but this is not a crucial parameter since arbitrarily large constant failure probability is already achievable just from sampling according to the row norms of $\mathbf{V}$). Critically, we achieve the same error bound using only row norm sampling from $\mathbf{V}$. This observation was missed in [1] and is essential since computing the column norms of $\mathbf{A}$ is expensive.  [1] instead tries to estimate each of the column norms of $\mathbf{A}$ by uniform sampling, which in general is not possible, even under the assumptions we make in our paper, as it is easy to miss large entries. We note that our assumptions are used instead only to estimate the row norms of $\mathbf{A}$, as we describe next.
>
> - Another main issue with [1] is that they need to estimate the row norms of $\mathbf{A}$ in order to compute $\mathbf{D^{-1}}$ to normalize the rows. [1] gives a heuristic for doing this based on interpolating the average value of the unread entries in a row based on the corresponding entries in the row in the columns of $\mathbf{A}$ that they sample, but there are examples where this heuristic fails, since heavy entries may be missed by this interpolation. In short, [1] provides no end-to-end theorems unlike our Theorem 1 and Corollaries 1 and 2, which we consider to be the main theoretical contributions of our work.
>
> - We would like to stress that the difference between our work and [1] is not only in the theoretical justification, but the algorithms themselves are also different in fundamental ways: (1) for finding the subset of columns of $\mathbf{D^{-1} A}$ to use, we use only row-norm sampling from $\mathbf{V}$ whereas [1] also uses column norm sampling from $\mathbf{A}$, and (2) although our work and [1] use uniform sampling of entries in $\mathbf{A}$, [1] uses this for the purpose of finding the subset of columns of $\mathbf{A}$ to use (and has no end-to-end guarantees for this since uniform sampling of columns is not guaranteed to estimate the column norms of A without strong assumptions), whereas we use uniform sampling only for the purpose of estimating the *row norms* of $\mathbf{A}$, so that we can create the normalization matrix $\mathbf{D}^{-1}$ (and [1] only provides a heuristic for estimating entries of $\mathbf{D}^{-1}$ and does not argue that they accurately estimate such entries).

---

### Meta-Review · Area_Chair_H1E4 · 2023-12-07

**Metareview:**

the paper provides a fast method (w.r.t the sequence length) for computing an attention layer. The algorithm comes with theoretical guarantees (under some assumptions) based on sketching techniques. The approach is agreed by the reviews to be novel even though there are quite a few papers tackling the exact same problem of fast attention layer computation. The main issue raised regarded the experiments, where the reviews raised (1) some missing baselines and (2) a need for a better empirical analysis of the runtime and hyper-parameter selection. During the rebuttal the authors provided a new experiment that alleviated the concerns regarding the first point. The second issue remained a concern for one reviewer.
Although there may be room for improvement in terms of the empirical analysis, the main issue in my opinion seems to be resolved as the theoretical guarantees are validated empirically. Adding to that, the paper brings a novel approach for an important well-explored problem, leading me to recommend accepting the paper. I think it will be of interest to the ICLR community.

**Justification For Why Not Higher Score:**

The empirical analysis has room for improvement

**Justification For Why Not Lower Score:**

The paper provides a novel and convincing method to an established problem

---

### Decision · Program_Chairs · 2024-01-16

Accept (poster)